# Homologous *bd* oxidases share the same architecture but differ in mechanism

Alexander Theßeling[1,4], Tim Rasmussen[2,4], Sabrina Burschel[1], Daniel Wohlwend [1], Jan Kägi[1], Rolf Müller[3], Bettina Böttcher [2]* & Thorsten Friedrich [1]*

Cytochrome *bd* oxidases are terminal reductases of bacterial and archaeal respiratory chains. The enzyme couples the oxidation of ubiquinol or menaquinol with the reduction of dioxygen to water, thus contributing to the generation of the protonmotive force. Here, we determine the structure of the *Escherichia coli bd* oxidase treated with the specific inhibitor aurachin by cryo-electron microscopy (cryo-EM). The major subunits CydA and CydB are related by a pseudo two fold symmetry. The heme b and d cofactors are found in CydA, while ubiquinone-8 is bound at the homologous positions in CydB to stabilize its structure. The architecture of the *E. coli* enzyme is highly similar to that of *Geobacillus thermodenitrificans*, however, the positions of heme $b_{595}$ and d are interchanged, and a common oxygen channel is blocked by a fourth subunit and substituted by a more narrow, alternative channel. Thus, with the same overall fold, the homologous enzymes exhibit a different mechanism.

[1] Institut für Biochemie, Albert-Ludwigs-Universität, Freiburg, Germany. [2] Biocenter and Rudolf Virchow Center, Julius-Maximilians-Universität, Würzburg, Germany. [3] Department of Microbial Natural Products, Helmholtz Institute for Pharmaceutical Research Saarland, Helmholtz Centre for Infection Research and Department of Pharmacy at Saarland University, Saarbrücken, Germany. [4] These authors contributed equally: Alexander Theßeling, Tim Rasmussen. *email: Bettina.Boettcher@uni-würzburg.de; Friedrich@bio.chemie.uni-freiburg.de

Oxygenic photosynthesis led to the rise of dioxygen, a strong oxidizing agent, in the atmosphere[1,2]. The use of dioxygen as terminal electron acceptor in respiratory chains enables cells to harvest a maximum amount of redox energy to generate a protonmotive force (pmf)[3]. However, the use of dioxygen may lead to the formation of partly reduced species that are highly cytotoxic and explains the need for sophisticated enzyme mechanisms catalyzing the four-electron reduction of dioxygen to water. Membrane-integrated oxidases use dioxygen as terminal electron acceptor[4,5]. They include the family of cytochrome *bd* oxidases that are terminal reductases of bacterial and archaeal respiratory chains[4,5]. The *bd* oxidases couple quinol oxidation and release of protons to the periplasmic side with proton uptake from the cytoplasmic side to reduce dioxygen to water. Thus, the *bd* oxidases contribute to the pmf by a vectorial charge transfer[4,5]. They display a high affinity toward dioxygen enabling growth under micro-aerobic conditions. In addition, they are required for a rapid dissociation of gaseous inhibitors, such as nitric oxide and enable growth under oxidative stress conditions in the presence of, e.g., $H_2O_2$[6,7]. However, the mechanism of *bd* oxidases is still under debate due to lack of the structure of the *Escherichia coli bd*-I oxidase, one of the two *bd* oxidases in this bacterium that has been intensively used for most biochemical and biophysical characterizations[8].

*E. coli bd*-I oxidase (called *bd* oxidase for simplicity hereafter) is supposed to consist of two major subunits named CydA and CydB and a small third subunit called CydX. CydA and CydB contain 9 transmembrane (TM) helices each, while CydX is made up of a single-TM helix. CydA harbors all three heme groups of the oxidase, namely the low-spin heme $b_{558}$, serving as electron input device and catalyzing quinol oxidation, the high-spin heme $b_{595}$ that is expected to deliver electrons to the third, the unique d-type heme, which is the site where dioxygen is reduced to water[9–11]. CydA contains a soluble, periplasmic domain, the Q-loop that varies in length leading to a "long" Q-loop (e.g., in *E. coli*) or a "short" Q-loop (e.g., in *Geobacillus thermodenitrificans*). This domain is expected to be involved in quinone binding and oxidation[5,11,12]. Recently, the structure of the *bd* oxidase from *G. thermodenitrificans* was obtained at 3.1–4 Å resolution by x-ray crystallography[13]. Here, we report the structure of the *bd* oxidase from *E. coli* at 3.3 Å resolution obtained by cryo-electron microscopy (cryo-EM). Our results suggest a different mechanism of the homologous enzymes and might furthermore provide important indications on the development of new antimicrobials that target the *bd* oxidase of pathogens such as Mycobacteria[14,15].

## Results

**Structure of *E. coli bd* oxidase.** The structure of the *E. coli bd* oxidase in the presence of the specific inhibitor aurachin C was obtained at 3.3 Å resolution by cryo-EM. Details on the cryo-EM procedure, the obtained local resolution and the cryo-EM parameters and statistics are provided in Supplementary Figs. 1 and 2 and in Supplementary Table 1. We were not able to obtain a structure of the *E. coli bd* oxidase in the absence of the inhibitor although aurachin C is not detectable in the electron density (see below). The two major subunits of the enzyme, CydA and CydB, share the same fold consisting of 9 TM helices that are arranged as two four-helix bundles and one additional peripheral helix (Fig. 1; Supplementary Fig. 3). The TM domains of both subunits superpose well when applying a twofold rotational symmetry revealing an rmsd of only 4.3 Å. This suggests that *cydA* and *cydB* arose by gene duplication although this is not reflected in the primary sequences (Supplementary Fig. 4). The two subunits interact via hydrophobic residues at the contact sites provided by TM helices 2, 3, and 9 of CydA and the symmetry-related helices

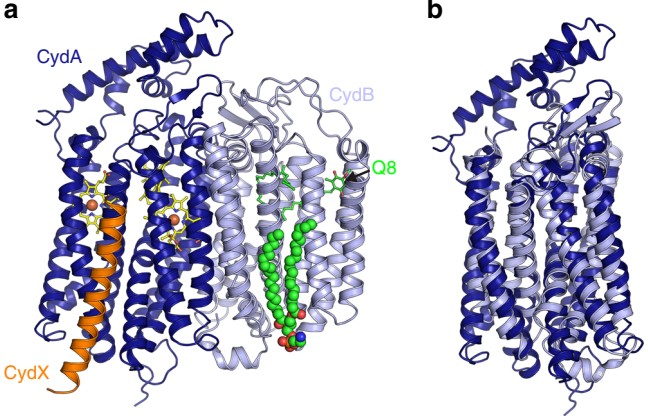

**Fig. 1** Cryo-EM structure of *E. coli bd* oxidase at 3.3 Å resolution. **a** Subunits CydA (dark blue) and CydB (light blue) comprise nine TM helices each. CydX (orange) is a single-TM helix, serving as clamp for the four-helix bundles coordinating hemes $b_{558}$ and $b_{595}$ (shown as sticks). Ubiquinone-8 (Q8) and a glycerophospholipid (shown in spheres) are bound to CydB. **b** CydA and CydB share the same fold, highlighting their homology. The extended lid of the Q binding site at the periplasmic side of the membrane (top), the Q-loop, is unique for CydA

on CydB. The Q-loop of CydA is located on the periplasmic side between TM helices 6 and 7 (Fig. 1; Supplementary Fig. 3) extending over a length of 136 amino acids including the short linkers to the helices (Supplementary Fig. 4). The defined part consists of 5 short helices connected by short loops that form a lid on top of CydA. 40 amino acids of the C-terminus of the Q-loop (positions 262–302) are not resolved in the structure, most likely due to high flexibility, suggesting a hinge function of this region for the lid (Fig. 1). The C-terminus of CydB forms a small cytoplasmic helix parallel to the membrane plane. In addition, a glycerophospholipid is bound to CydB (Fig. 1). It consists of two C-18 fatty acids as revealed in the structure and ethanolamine as polar head group as determined by TLC analysis (Supplementary Fig. 5) that is held in place by an electrostatic interaction with Arg365. Interestingly, Arg191 is placed just above Arg365 and could probably substitute its function, when binding phospholipids with shorter chain lengths (Supplementary Fig. 5). Both arginine residues are not conserved. The alkyl chains interact with the TM helices 1, 2, 5, 8, and 9 of CydB by several hydrophobic interactions.

The single helix subunit CydX is made up of 37 residues and interacts with TM helices 1, 5, and 6 of CydA in a stretched conformation (Fig. 1). The C-terminal part of CydX contains mainly hydrophilic amino acids and is exposed to the cytoplasm with Glu25 as a "float" preventing the helix to fully enter the membrane. The N-terminal part of CydX resides within the membrane. Its multiple interactions with CydA might explain that the oxidase is not properly assembled in a Δ*cydX* strain[16]. Unexpectedly, the *E. coli bd* oxidase contains a fourth subunit that modeled as a single helical protein with an expected molecular mass of about 3 kDa as deduced from a silver-stained sodium dodecyl sulfate polyacrylamide gel electrophoresis of our preparation (Supplementary Fig. 6). The molecular identity of this subunit was determined by nano-liquid chromatography electrospray ionization mass spectrometry/MS to YnhF, an *E. coli* protein with one predicted TM helix and located in the cytoplasmic membrane (Supplementary Fig. 6)[17–20]. The protein is produced in conditions of low oxygen concentration[17,19] and has retained its N-terminal formyl-methionine[18]. The corresponding gene is not part of the *cyd*-operon encoding the *E. coli bd* oxidase just consisting of *cydA, B,* and *X*[16], but is located

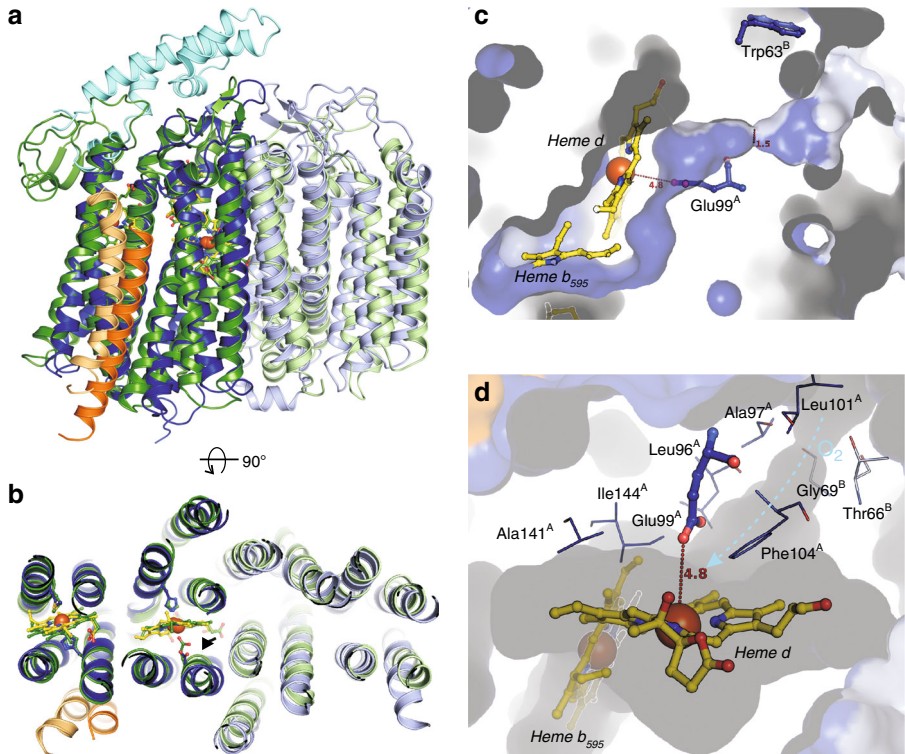

**Fig. 2** The heme arrangement in *E. coli bd* oxidase provides a substrate cavity at heme d. **a** In comparison with the *G. thermodenitrificans* enzyme (pdb ID 5DOQ, CydA: dark green, CydB: light green, CydS: light orange), the *E. coli* enzyme features a positional exchange of hemes b[595] and d. **b** As opposed to b[595] of *G. thermodenitrificans bd* oxidase, the central iron of heme d in the *E. coli* enzyme does not recruit Glu99 of CydA as axial ligand, leaving a voluminous cavity for substrates (arrow). **c** CydA (dark blue surface) and CydB (light blue surface) form a channel from the solvent to heme d with maximum constriction to 1.5 Å. The dislocation of Glu99 (CydA) provides space for a small substrate like dioxygen to bind as axial ligand to heme d. **d** Several hydrophobic residues surround Glu99[A], forming a hydrophobic roof above the cavity harboring heme d. The putative oxygen channel shown in (**c**) is indicated with an ice blue arrow. The constriction of this channel is formed by hydrophobic moieties of both CydA (Ala97 and Leu101) and CydB (Thr66 and Gly69)

between *ydhP* encoding a putative transporter and the transcriptional repressor gene *purR*[17,18]. We propose to rename the protein to CydY. Noteworthy, a TM helix with stretches homologous to YnhF is also found in several membrane proteins containing more than one TM helix, including eukaryotic membrane proteins. In *E. coli*, this fourth subunit, CydY, seems to play a crucial role in the mechanism of the *bd* oxidase (see below).

**Comparison to the *G. thermodenitrificans* enzyme.** Unexpectedly[21], the overall structure of the *E. coli* oxidase is highly similar to that reported for the *G. thermodenitrificans* enzyme (Fig. 2)[13]. Remarkably, there are small but substantial differences that presumably result in different mechanisms. Surprisingly, although similarly positioned (Fig. 2)[13,22], the relative arrangement of the three heme groups in CydA is strikingly different (Fig. 3). They form a triangle with consistent distances between their central iron atoms and they show the same orientation of the heme planes relative to the membrane plane (Figs. 2 and 3). The low-spin b[558] is unambiguously identified due to the hexa-coordination of the central iron by His186 and Met393 and it is positioned below the "lid" provided by the long Q-loop (Fig. 1) as expected from mutagenesis and spectroscopy studies (Fig. 2)[8]. Heme d, however, found closer to the periplasmic side in the *G. thermodenitrificans* structure, and heme b[595], oriented to the middle of the membrane[13], occupy interchanged positions in *E. coli* as demonstrated by significantly better density fits (Figs. 2 and 3). The proposed heme arrangement is in line with the spectroscopic characterization of *E. coli* variants, in which Glu99

and Glu445, the axial ligands to heme b[595] and d, have been mutated[8,23–26]. In our model, heme b[595] is closer to the periplasmic space and ligated by Glu445 and heme d itself is positioned more to the middle of the membrane with Glu99 as axial ligand. This is backed by the finding that mutating Glu445 indeed had the strongest effects on the ultraviolet (UV)–vis spectra of hemes b[595] and b[558][8]. Further on, it was shown that *E. coli* Glu445 is critical for the electrochemical properties of heme b[595][12,23]. Mutations of Glu99 led to either the loss of the heme b[595]/heme d active site[24–26,33], or solely to the loss of heme d[8,26] implying at least a proximity of Glu99 to heme d. We changed the non-conserved Glu74[A] located at the surface to CydB to a phenylalanine residue. The residue is located at the level of heme d. The sequences of the mutagenic primers are listed in Supplementary Table 2. The E74F[A] variant was instable and could not be purified but the absorbances for heme d were lacking in the redox difference of the detergent extract, while a residual absorbance of b[595] is clearly detectable supporting our proposed model (Supplementary Fig. 7).

In our model, Glu99 is an axial ligand of heme d. The homolog in *G. thermodenitrificans* Glu101 has a very short distance of only 2.1 Å to the central Fe of the positionally equivalent heme b[595]. In *E. coli* by contrast, the corresponding Glu99 faces the central Fe of heme d in an axial distance of 4.8 Å, providing space for substrates. The displacement is due to the insertion of Leu101 into TM helix 3 resulting in a stronger curvature (Fig. 2). This provides a voluminous cavity for potential substrates to bind at the axial position of heme d, roofed by the hydrophobic Ile144 and Phe104. In fact, dioxygen would fit well into that cavity. This

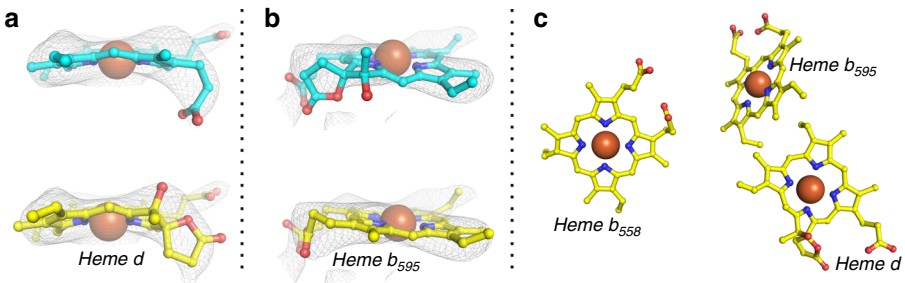

**Fig. 3** Heme arrangement in CydA. Electron density maps of $b_{595}$ and d contoured at 1.5$\sigma$ show that, as compared to *G. thermodenitrificans bd* oxidase, $b_{595}$ and d exchange positions in CydA (**a**, top). Heme $b_{595}$ as found in *G. thermodenitrificans bd* oxidase (blue carbons) does not fit well into the electron density map, neither does heme d (**b** top). Exchanging positions of both heme groups (yellow carbons) produces substantially better fits of hemes d (**a**, bottom) and $b_{595}$ (**b**, bottom), in particular for the hydroxyl group at the spiro substituted pyrrole of heme d. **c** Arrangement of heme groups in *E. coli bd* oxidase, periplasm: top of picture, cytoplasm: bottom of picture. The proposed arrangement is in line with the spectroscopic characterization of *E. coli bd* variants, in which Glu99$^A$ and Glu445$^A$, both adjacent to the central Fe ion of heme $b_{595}$ and d, respectively, have been mutated[8,23–26]. Mutation of Glu99$^A$, adjacent to heme d according to our model, did not influence the heme $b_{595}$ content, while mutations of Glu445$^A$, adjacent to heme $b_{595}$ according to our model, either influenced solely the content of $b_{595}$ or of both hemes in question[8,23–26]. Mutation of Glu74$^A$, close to heme d according to the proposed model, eliminated the signal of heme d (Supplementary Fig. 7)

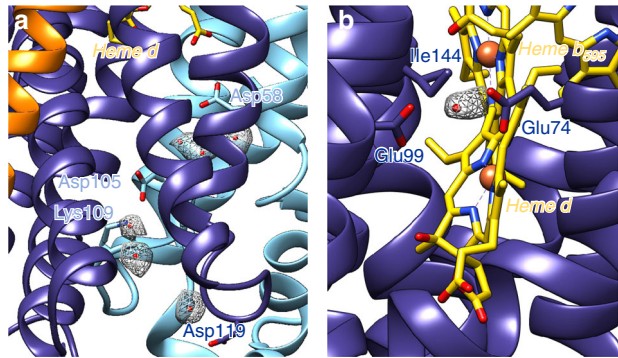

**Fig. 4** Water densities within *bd* oxidase. **a** A chain of water molecules at the subunit interface was identified providing a proton path from the cytosol to Asp58 on CydB. Protons can be transferred from this residue to the propionate of heme d. **b** A density between Glu99 and Glu74 close to heme d could be caused by the substrate dioxygen or the product water. CydA is shown in dark blue, CydB in light blue and CydX in orange. Densities of water are shown as meshes

finds support in an extra density that is located in this cavity between E99 and E74, probably caused by a dioxygen or a water molecule (Fig. 4). Further, it was proposed that heme d is derived from bound heme b during enzymatic turnover by hydroxylation of the porphyrin ring[8], requiring protons. However, our model suggests that the proton transfer between Glu99 and Glu445 needed for porphyrin hydroxylation is in fact disrupted (see below). Hence, the b heme coordinated by Glu99 is the sole candidate for hydroxylation in *E. coli bd* oxidase, possibly explaining the different cofactor arrangement.

While heme $b_{558}$ is spatially more separated from hemes $b_{595}$ and d with Fe–Fe distances of 18.6 and 15.2 Å, the Fe–Fe distance between hemes $b_{595}$ and d is only 11.1 Å, in line with previous observations[13,27]. In fact, the shortest edge-to-edge distance between the latter two is only 3.5 Å (Fig. 3) explaining their cooperativity[28]. Trp441 proposed as being liable for donating an electron to the formation of a reaction intermediate, the F+$^*$ state[29], is found in conserved position in *E. coli*.

**CydB contains a tightly bound ubiquinone-8**. We found a large electron density extending along hydrophobic clefts in CydB (Fig. 5), perfectly fitting to ubiquinone-8 (Q-8). The quinone

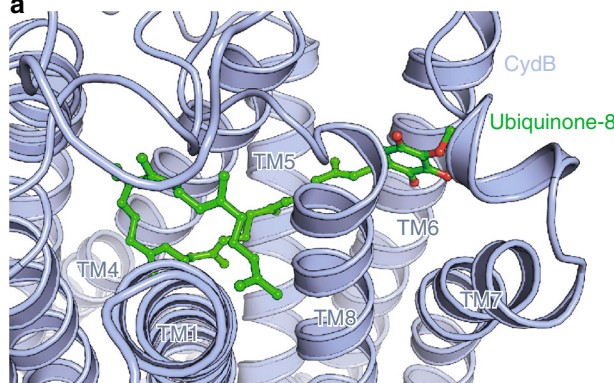

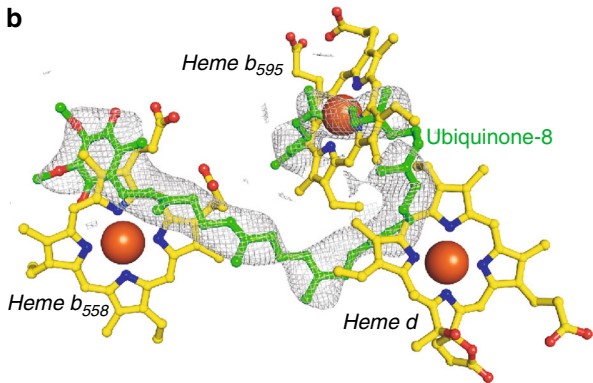

**Fig. 5** *E. coli bd* oxidase features a Q-8 binding site in CydB that has a stabilizing function. **a** Ubiquinone-8 (shown as sticks) binds to subunit CydB (light blue). It deeply inserts into CydB, involving six TM helices in binding. **b** A superposition of CydA and CydB shows that in CydB, ubiquinone-8 follows the trace of the heme positions in CydA

head group points away from CydA into the membrane space, indicating a structural rather than a functional role of Q-8 (Fig. 5). The overlay of CydA and CydB shows that Q-8 follows the arrangement of the heme groups in CydA. We examined the role of bound Q-8 by mutating conserved alanine residues at the surface of the hydrophobic cleft to phenylalanine residues that might disturb binding of Q-8. The membranes from the A82F$^B$, A137F$^B$, and the A172F$^B$ mutants showed 80 to 70% of the

NADH oxidase activity of the membranes of the parental strain. We attributed the lower activity to a hindered Q-8 binding supporting the idea that bound Q-8 stabilizes CydB by substituting the heme groups. Remarkably, the excess aurachin C did not substitute the Q-8 molecule indicating its tight binding.

**Binding of aurachin**. Aurachins prevent the binding of the quinol substrate to the oxidases[30–32]. We determined the apparent $IC_{50}$ of aurachins C and D towards the duroquinol:dioxygen oxidoreductase activity of our preparation to 12 and 35 nM, respectively, emphasizing its specific binding (Supplementary Fig. 7). Due to its high affinity towards the *bd* oxidase, we incubated the preparation with 1 µM aurachin C prior to cryo-EM. Although we obtained the structure of the *E. coli bd* oxidase only in the presence of aurachin C, we unexpectedly found only little density assignable to aurachin at the proposed quinol binding site, including Glu257 and Lys252 of CydA[5,11,12,33]. Most likely, the flexibility of the Q-loop above the binding site prohibits a reliable assignment.

**The oxygen entry site**. Heme $b_{595}$ is directly connected to the protein surface by a solvent accessible substrate channel also identified in the *G. thermodenitrificans* enzyme (Supplementary Fig. 8)[13]. Noteworthy, in *E. coli* the cavity extends to $b_{595}$ but not to heme d that binds dioxygen. A deeper penetration of dioxygen into that channel towards heme d is blocked by heme d itself. Thus, as dioxygen has to bind to heme d and not to $b_{595}$ the conserved access to the latter has to be inaccessible in *E. coli* oxidase. Indeed, a density is detectable at this position that modeled as a single helical structure and that was identified as CydY (Supplementary Fig. 6). It is proposed that CydY is a fourth subunit of the *E. coli bd* oxidase blocking the dioxygen channel present in both enzymes. An alternative pathway to guide dioxygen to heme d not present in the *G. thermodenitrificans* enzyme is provided by a small hydrophobic channel starting above Trp63 on CydB and extending further to heme d on CydA (Fig. 2). The *E. coli* specific channel has a constriction with a diameter of about 1.5 Å that might act as a selectivity filter. While dioxygen might easily pass through the constriction, angled molecules such as hydrogen sulfide may not.

**Proton pathways**. The reaction of the *bd* oxidase is coupled with the generation of a pmf due to proton uptake from the cytoplasmic side to reduce dioxygen to water while protons released by quinol oxidation are delivered to the periplasmic side. Several proton pathways from the cytoplasmic side to the active site have been proposed based on the structure of the *G. thermodenitrificans* enzyme and by mutagenesis studies using the *E. coli* enzyme[8,13,23,24,33,34]. Our structure shows a water chain along a hydrophilic channel for proton transfer, entering the enzyme at $D119^A$, along $K57^A$, $K109^B$, $D105^B$, $Y379^B$ to $D58^B$ (Fig. 4). From here protons can be transferred to the propionate group of heme d. In *G. thermodenitrificans*, the proton has to be transferred further to the homolog of *E. coli* Glu445, the ligand of heme d, potentially involving the heme d propionate. This proton transfer is no longer required in the *E. coli bd* oxidase and accordingly blocked by Ile144 that acts as a hydrophobic separator.

## Discussion

The *bd* oxidases from *E. coli* and *G. thermodenitrificans* are homologous enzymes (Supplementary Fig. 4) and adopt a highly similar architecture (Fig. 2). However, the position of the hemes $b_{595}$ and d are interchanged leading to a different position of the binding site for the gaseous substrates (Figs. 2 and 3). One spacious entry site for the gaseous substrates is conserved between the two enzymes, however, blocked by a fourth subunit, CydY, in the *E. coli* enzyme (Supplementary Fig. 6). The second binding site only present in the *E. coli* enzyme is accessible by a long channel that contains a constriction possibly conferring substrate specificity. Together with the low activity of the *G. thermodenitrificans*[13] enzyme as compared to the *E. coli* enzyme with high turnover[35] this indicates that the former enzyme is predominantly involved in the detoxification of the bacterium's environment, while the latter seems to act as a true oxidase. The presence of *cydY* (or formerly *ynhF*) encoding the fourth subunit might therefore be used as a marker to identify the molecular tasks of the homologous enzymes with different mechanisms.

## Methods

**Preparation of *bd* oxidase from *E. coli* BL21\* Δ*cyo*/pET28a *cydA_hBX***. An *E. coli* strain lacking the chromosomal copy of the *cyo* genes encoding $bo_3$ oxidase was used. The strain was transformed with pET28a $cydA_hBX$[16]. All steps to purify *bd* oxidase were carried out at 4 °C. Frozen cells (35–40 g wet weight) were suspended in the 6-fold volume 20 mM 3-(N-morpholino)propanesulfonic acid (MOPS), 20 mM NaCl, 1 mM phenylmethylsulfonyl fluoride (PMSF), pH 7.0 and disrupted by single pass through a French pressure cell (SLM Aminco) at 16,000 psi. Cell debris were removed by centrifugation at $12,000 \times g$ for 20 min. Cytoplasmic membranes were sedimented by ultracentrifugation of the cleared lysate at $250,000 \times g$ for 75 min and suspended in 20 mM MOPS, 20 mM NaCl, 0.5 mM PMSF, pH 7.0 to a final protein concentration of 10 mg/mL. Membrane proteins were solubilized by incubating the membrane suspension 1 h with 1% Lauryl Maltose Neopentyl Glycol (BioFroxx) stirring at 4 °C. Nonsolubilized material was separated by ultracentrifugation at $250,000 \times g$ for 15 min. The supernatant was applied to a 25 ml HisTrapFF (GE Healthcare) column pre-equilibrated in 50 mM MOPS, 500 mM NaCl, 50 mM imidazole, 0.5 mM PMSF, 0.003% MNG, pH 7.0, with a flow rate of 2 mL/min. The column was washed with buffer containing 68 mM imidazole until the UV-baseline dropped to the original value. Bound proteins were eluted in a linear gradient of 3 CV from 68 to 500 mM imidazole. Peak fractions were pooled and washed two times with buffer (20 mM MOPS, 20 mM NaCl, 0.5 mM PMSF, pH 7.0). The concentrated sample was then applied onto a MonoQ 10/100 GL ion exchange column (GE Healthcare) in 20 mM MOPS, 20 mM NaCl, 0.5 mM PMSF, pH 7.0 and eluted in a gradient of 10 CV up to 350 mM NaCl. The concentrated sample was then applied onto a Superose 6 Inc 10/300 GL (GE Healthcare) size-exclusion chromatography column equilibrated in buffer (20 mM MOPS, 20 mM NaCl, 0.5 mM PMSF, pH 7.0) at a flow rate of 0.2 mL/min. Peak fractions were pooled and used for further analysis. For cryo-EM the samples were treated with 1 µM aurachin C.

**Cryo-electron microscopy and data analysis**. 3.5 µl of 2 mg/mL *bd* oxidase in amphipol A8–35 was applied to a glow-discharged, holey carbon coated copper 400 grid R1.2/1.3 (Quantifoil) and vitrified in liquid ethane with a Vitrobot mark IV (FEI) with following settings: 3.5 s blot time, 5 blot force, 0 s drain and waiting time, 100% humidity at 4 °C. Data were collected at the Würzburg cryo-EM facility on a Titan Krios G3 (Thermo Scientific) equipped with a Falcon III direct detector. The magnification was set to 75,000 (calibrated pixel size 1.0635 Å/pixel), the condenser aperture to 70 µm, and the objective aperture to 100 µm. Movies with 47 fractions were recorded in counting mode at parallel illumination with a total exposure of 59 $e^-/Å^2$. A defocus range of 1.4–2.2 µm was applied. Movies were motion corrected and dose weighted with the program MotionCor2[36] and the contrast transfer functions were obtained with the program CTFFind4[37]. The data were further processed with the program Relion 3.0[38,39]. Manual particle picking and 2D classification provided templates for auto-picking in Relion. The particle set was cleaned up with two rounds of 2D classification. An initial model was obtained with the InitialModel option of relion_refine and thereafter used for the auto-refinement. Two rounds of 3D classification provided the final particle set which was used for a final auto-refinement. The local resolution was estimated with the locres option or relion_postprocess. The program Coot 0.8.9.1[40] was used for model building using the crystal structure of *G. thermodenitrificans bd* oxidase (pdb: 5DOQ) as starting model. The model was refined with the program Phenix realspace.refine version 1.13.2998[41,42] and validated with MolProbity 4.4[43]. Images were prepared with the programs Chimera 1.11[44] and PyMOL 1.8.

**Determination of the polar head group of bound phospholipids**. 500 µL of a *E. coli bd* oxidase preparation (20 mg/mL) was mixed with 1 mL chloroform/methanol (2:1, v/v) and incubated at RT for 45 min. 200 µL pure water was added and the suspension was mixed again. After centrifugation (1 min, RT, 1000 rcf) the organic phase was separated and evaporated in a nitrogen flow. The residual was dissolved in 100 µL chloroform. 5 µL of the concentrated organic phase and of

standard solutions of phosphatidylglycerol (0.013 M), phosphatidylethanolamine (0.015 M), phosphatidylserine (0.012 M) and phosphatidylcholine (0.032 M, all from Avanti, Alabaster, Alabama) were placed on the TLC plate (silica gel 60 F254, 20 × 20 cm, Merck, Darmstadt, and dried for 5 min at 60 °C. The plate was placed into a TLC chamber, filled with chloroform–methanol–water (75:25:2.5, v/v/v) as mobile phase. The separation was stopped when the front of the mobile phase was 0.5 cm below the end of the plate. The plate was dried in a drying cabinet for 5 min at 60 °C. TLC plates were developed by spraying with primuline solution and drying for 5 min at 60 °C. Phosphorylated TLC-spots were detected with a Fluorescence imager (Fusion SL, Vilber Lourmat) using a Blue535Y filter and 80 ms detection time.

**Amphipol exchange.** To replace the detergent MNG, isolated *bd* oxidase was supplemented with a threefold mass excess of amphipol A8–35 (Anatrace, Maumee, Ohio). The sample was incubated at 4 °C for 90 min while stirring. The excess detergent was removed by addition of 40 fold mass excess BioBeads (SM-2 resin, BioRad) and further 3 h incubation at 4 °C under mild stirring. To remove excess amphipols, the sample was loaded onto a Superose 6 Increase (24 mL) size-exclusion column equilibrated in 20 mM MOPS, pH 7.0, 20 mM NaCl. The homogenous peak fractions contained the enzyme in amphipol A8–35 without detergent.

**Inhibition of *bd* oxidase by aurachin C and D.** The inhibitory potency of aurachin C and D on the isolated *E. coli bd* oxidase was determined by measuring the duroquinol:oxygen oxidoreductase activity[35]. 10 µL of a 1 M DTT solution prepared under anoxic conditions were injected in the chamber of an oxygen electrode (oxygraph+, Hansatech) in 2 mL buffer (20 mM MOPS pH 7.0, 20 mM NaCl, 0.003% MNG) at 30 °C. 5 µL of a 100 mM of duroquinone (Sigma) in ethanol were added. The reaction was started by an addition of 1 µl *bd* oxidase (1 mg/mL). Various amounts of aurachins C and D were added to the reaction. The non-enzymatic reaction was less than 1% of the rate of the enzymatic reaction. The enzymatic rate was corrected for the non-enzymatic one. Each data point was assayed in triplicates.

**Reporting summary.** Further information on research design is available in the Nature Research Reporting Summary linked to this article.

## Data availability

The cryo-EM maps have been deposited in the Electron Microscopy Data Bank (EMDB) under the accession number EMD-10049. The atomic model has been deposited in the Protein Data Bank under accession number 6RX4. The source data underlying Supplementary Figs. 5b, 6b 7a, 7b, and 9d are provided as a Source Data file. Other data are available from the corresponding authors upon reasonable request.

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

## Acknowledgements

We thank Drs. Schara Safarian and Alexander Hahn, Frankfurt, for most helpful discussions and sharing unpublished data. Electron microscopic data were acquired at the cryo-EM facility of the University Würzburg (DFG equipment grant INST 93/903-1 FUGG). This work was supported by the Deutsche Forschungsgemeinschaft by grants—278002225/RTG 2202 and 235777276/RTG 1976.

## Author contributions

A.T. and J.K. prepared the enzyme and performed the amphipol exchange, A.T. performed enzyme kinetic measurements and determined the identity of the lipid, T.R. and A.T. made the EM grids, T.R. and B.B. measured the samples, solved the structure, and refined the model, S.B. made and grew mutants, A.T. and S.B. recorded UV–vis spectra, R.M. provided the aurachins, D.W., T.R., A.T., and B.B. made the figures, T.F. designed the project and wrote the paper with corrections from all co-authors.

## Competing interests

The authors declare no competing interests.
