## [Peer Review File · Nature Communications]

Reviewers' comments:

Reviewer #1 (Remarks to the Author):

The paper by Thiesseling et al. describes the structure of cytochrome bd from *E. coli* solved by cryo-EM. This study complements, but also significantly extends previous work on the homologous enzyme from *G. thermodenitrificans* (Safarian et al. 2016, cited as reference 12). The *E. coli* structure shows very important new features such as tightly bound ubiquinone and lipid molecules and a new subunit. Highly puzzling, the arrangement of the heme cofactors is inverted, requiring an alternative oxygen access channel.

Several points of the manuscript, in particular regarding presentation, should be addressed before publication can be considered.

1. Which inhibitor was used to obtain the structure? According to Results section, aurachin C (line 66, 68), but the Methods section says aurachin D was used for activity assays (line 333). Please clarify. The type and concentration of aurachin used for the structure determination should be stated in the Methods section.

2. Line 74 – 82, section on Q-loop

According to main text, this loop consists of 146 residues. In the Q-loop alignment (Supplementary Figure 2, panel B) clearly more residues are shown. Which are the residues of the Q-loop? Which residues are regarded as linkers to the (transmembrane?) helices?

The authors argue that part of the Q-loop is resolved, as shown in Fig 1, whereas the non-resolved part probably is flexible and may act as hinge. With the current presentation it is difficult to evaluate which regions/residues of this loop are resolved and which are flexible. A separate supplementary figure on the Q-loop may exceed the scope of this paper, but the authors could either describe in main text or label in the alignment of Supplementary Fig 2/panel B which residues are resolved, which are regarded as linkers and which apparently are flexible.

3. Supplementary Figure 2: the letters used are very small, even after zooming in on screen. Please use larger font and clarify in legend what the "." symbol means in the alignment as opposed to ":"

4. Line 85, importance of arginine residues for binding of the lipid molecule. Are these arginine residues conserved?

5. Line 127: The roof of the potential O₂ binding site at heme D is formed by Ile44 according to main text, but Fig 2D shows Ile144. Which is it?

6. Access channel for O₂: the main text refers to CydB Trp63 as entry point of the suggested access channel. Can this residue be indicated in a figure, e.g. Fig 2C or D?

7. The authors may consider presenting the figure on water channels, which now is Supplementary Figure 5, as main figure, possibly including the "stop residue" Ile144 (if this does not exceed the max. number of allowed display items).

8. Are Supplementary Figures 7 and 8 somewhere referred to in main text? Please check.

9. References

Please check for clarity regarding references/citations:

Sentence from line 50: Not all three references (9-11) deal with heme b558. Please cite one reference per heme group or move citation to end of sentence.

Line 56: References 9 and 10 do not deal with Q-loop. Cite only reference 11?

Is reference 10 the same as reference 20?

9. Textual: please check for typos, e.g. line 20 "contribute to" and line 27 "noarrow", line 31 ("enables"), line 47 ("is" is missing).

Reviewer #2 (Remarks to the Author):

In the manuscript entitled "Homologous bd oxidases: same architecture, differing mechanisms" from Theßeling et al. the authors describe their structure of E.coli bd oxidase solved by cryo-EM. The manuscript is generally well written, and the figures are precise and illustrative. The cryo-EM work is absolutely trustworthy, and given the comparably small size of this membrane protein, the provided resolution of 3.3 Å resolution is certainly adequate.

Please find my comments below:

In addition to a full structural description of the oxidase, the authors identified an additional subunit, modeled as an alpha helix of 3.5 kDa. It would undoubtedly be fascinating to know what this subunit is, especially as the authors attribute "a crucial role in the mechanism" to it. It should be possible to reveal the origin of this subunit using a mass-spec. approach and I wonder whether this is something the authors have considered?

The authors describe a rearrangement of the heme groups illustrated in Figure 3. From Figure 3 (and the corresponding caption) it appears as if this claim was made solely based on the quality of the fit into the density, which would be very risky. In the main text, the authors also reference spectroscopic experiments which should support their claim; however, the authors do not reveal in what way. I would suggest to already mention the supporting experiments in the figure caption as I do not think such a claim can already be made from the densities. In the main text, I would like to read (in summary) what the outcome of these five spectroscopic characterizations of the mutations was and why this supports the authors' interpretation.

In general, the corresponding section in the manuscript is worded too unspecific, which makes it difficult to follow the complex issue described. I would suggest to at least replace some of the "they/their etc." with actual definitions which would make it much easier for the reader to follow.

Reviewer #1 (Remarks to the Author):

The paper by Thiesseling et al. describes the structure of cytochrome bd from *E. coli* solved by cryo-EM. This study complements, but also significantly extends previous work on the homologous enzyme from *G. thermodenitrificans* (Safarian et al. 2016, cited as reference 12). The *E. coli* structure shows very important new features such as tightly bound ubiquinone and lipid molecules and a new subunit. Highly puzzling, the arrangement of the heme cofactors is inverted, requiring an alternative oxygen access channel.

A: We thank the reviewer for these very supportive comments.

Several points of the manuscript, in particular regarding presentation, should be addressed before publication can be considered.

1. Which inhibitor was used to obtain the structure? According to Results section, aurachin C (line 66, 68), but the Methods section says aurachin D was used for activity assays (line 333). Please clarify. The type and concentration of aurachin used for the structure determination should be stated in the Methods section.

A: We treated the samples for cryo-EM with aurachin C, which is stated on p. 4, l. 65 and 69. In the 'Online methods' section, p. 16, l. 319, we now state that 1 μ M aurachin C was used for structure determination. We have extended the section 'Binding of aurachin' in the main text p. 8., lines 169 to 178 because the apparent IC₅₀ values of aurachin C and D to bd-I oxidase were not known and we have determined both values. Clearly, we state in this section that aurachin C was used to determine the structure (p. 8, l. 174/175). The inhibition of bd-oxidase by aurachin C and D is now documented in a titration curve shown in the new Supplementary Figure 7.

2. Line 74 – 82, section on Q-loop

According to main text, this loop consists of 146 residues. In the Q-loop alignment (Supplementary Figure 2, panel B) clearly more residues are shown. Which are the residues of the Q-loop?

Which residues are regarded as linkers to the (transmembrane?) helices?

The authors argue that part of the Q-loop is resolved, as shown in Fig 1, whereas the non-resolved part probably is flexible and may act as hinge. With the current presentation it is difficult to evaluate which regions/residues of this loop are resolved and which are flexible. A separate supplementary figure on the Q-loop may exceed the scope of this paper, but the authors could either describe in main text or label in the alignment of Supplementary Fig 2/panel B which residues are resolved, which are regarded as linkers and which apparently are flexible.

A: We apologize for the typo: The length of the Q-loop is 136 amino acid residues. This is now stated on p.4, l. 78. The linker regions, the residues being part of the Q-loop and the residues not resolved in our structure are now unambiguously marked as such in Supplementary Figure 4 (Supplementary Figure 2 of our initial submission). The 40 residues that are not resolved in our structure (marked in red in Supplementary Figure 4) are expected to show flexibility (p. 5, l. 82 and legend to Supplementary Figure 4).

3. Supplementary Figure 2: the letters used are very small, even after zooming in on screen. Please use larger font and clarify in legend what the “.” symbol means in the alignment as opposed to “:”

A: The former Supplementary Figure 2 is Supplementary Figure 4 of the revised version. The figure is drawn completely new and with a larger font size to enable a better reading of the letters. All symbols used in the sequence comparisons are now explained in the legend to the figure.

4. Line 85, importance of arginine residues for binding of the lipid molecule. Are these arginine residues conserved?

A: We are very grateful for that important hint. The arginine residues that show electrostatic interactions with the polar headgroup of the phospholipids are not conserved. This is now stated on p. 5, l. 89.

5. Line 127: The roof of the potential O₂ binding site at heme D is formed by Ile44 according to main text, but Fig 2D shows Ile144. Which is it?

A: We are very sorry for causing confusion. It should be read Ile144. This has been corrected (p.7, l. 144).

6. Access channel for O₂: the main text refers to CydB Trp63 as entry point of the suggested access channel. Can this residue be indicated in a figure, e.g. Fig 2C or D?

A: The position of Trp63, the entry point of the novel dioxygen access channel, is now included in Fig. 2C.

7. The authors may consider presenting the figure on water channels, which now is Supplementary Figure 5, as main figure, possibly including the “stop residue” Ile144 (if this does not exceed the max. number of allowed display items).

A: According to the suggestion of the reviewer, we have included the former Supplementary Figure 5 as new Figure 4 in the main text. The position of the ‘stop residue’ Ile144 is now included in the figure.

8. Are Supplementary Figures 7 and 8 somewhere referred to in main text? Please check.

A: The former Supplementary Figures 7 and 8 were moved to Supplementary Figures 1 and 2 in the revised version. There is now a reference to these figures on p. 4, l. 67/68.

9. References

Please check for clarity regarding references/citations:

Sentence from line 50: Not all three references (9-11) deal with heme b558. Please cite one reference per heme group or move citation to end of sentence.

Line 56: References 9 and 10 do not deal with Q-loop. Cite only reference 11?

Is reference 10 the same as reference 20?

A: We are very grateful for this comment. There was a confusion with citing the references that has been corrected. The references not dealing with b_{558} were moved to the end of the sentence (p. 3, l. 53). References not dealing with the Q-loop have been replaced by other reference (our ref. 5 and the new reference 12; p. 4, l. 56). The former references 10 and 20 are not identical. By mistake the same title was used for both references (now refs. 10 and 25). This has been corrected.

9. Textual: please check for typos, e.g. line 20 “contribute to” and line 27 “noarrow”, line 31 (“enables”), line 47 (“is” is missing).

A: All corrections were made. We thank the reviewer for hinting at the typos. All authors have read and corrected the manuscript. So, we hope that there are “no” more typos in the text.

Reviewer #2 (Remarks to the Author):

In the manuscript entitled “Homologous bd oxidases: same architecture, differing mechanisms” from Theßeling et al. the authors describe their structure of E.coli bd oxidase solved by cryo-EM. The manuscript is generally well written, and the figures are precise and illustrative. The cryo-EM work is absolutely trustworthy, and given the comparably small size of this membrane protein, the provided resolution of 3.3 Å resolution is certainly adequate.

A: We thank the reviewer for these very supportive comments.

Please find my comments below:

In addition to a full structural description of the oxidase, the authors identified an additional subunit, modeled as an alpha helix of 3.5 kDa. It would undoubtedly be fascinating to know what this subunit is, especially as the authors attribute “a crucial role in the mechanism” to it. It should be possible to reveal the origin of this subunit using a mass-spec. approach and I wonder whether this is something the authors have considered?

A: We fully agree with the reviewer, but we had a hard time to identify the protein due to its small size. Now, we were successfully applying nano-LC-ESI-MS/MS to identify the additional subunit as the orphan protein YnhF. The function of this protein was not known but its features (one TM helix in the cytoplasmic membrane and an optimal expression of the corresponding gene under micro aerobic conditions) fit very nicely with our experimental data and its proposed function in bd oxidase. The protein is now described on p. 5/6, l. 99 to 109. The current knowledge about this protein is documented in the new references 17-20. The link between YnhF and bd oxidase was not known before; accordingly we propose to re-name the protein to CydY.

The authors describe a rearrangement of the heme groups illustrated in Figure 3. From Figure 3 (and the corresponding caption) it appears as if this claim was made solely based on the quality of the fit into the density, which would be very risky. In the main text, the authors also reference spectroscopic experiments which should support their claim; however, the authors do not reveal in what way. I would suggest to already mention the supporting experiments in the figure caption as I do not think such a claim can already be made from the densities. In the main text, I would like to

read (in summary) what the outcome of these five spectroscopic characterizations of the mutations was and why this supports the authors' interpretation.

In general, the corresponding section in the manuscript is worded too unspecific, which makes it difficult to follow the complex issue described. I would suggest to at least replace some of the "they/their etc." with actual definitions which would make it much easier for the reader to follow.

A: There are three lines of evidence to propose the re-arrangement of the heme groups in E. coli bd oxidase compared to the G. thermodenitrificans oxidase. First, our assignment is based on our structural data as pointed out by the reviewer. The heme arrangement as seen in the G. thermodenitrificans enzyme does not fit to the experimental densities. Conversely, the re-arranged heme groups fit very nicely. Second, our model is supported by data from mutagenesis studies described in the literature. Glu99 and Glu445 are the axial ligands of the two heme groups in question, namely heme b_{595} and heme d. In E. coli, mutation of Glu445 has the strongest influence on the amount and the spectroscopic and thermodynamic properties of heme b_{595} . In our model, Glu445 is a ligand of b_{595} . On the other hand, mutation of Glu99 influenced either both heme groups or just heme d alone. In our model, Glu99 is a ligand of heme d. This is now stated in the main text (p.6/7, l. 124 to 137). We were setting up a more direct text lacking most of the 'They/Their, etc.'. Finally, we introduced a new mutation on Glu74^A, not directly involved in heme binding but being located at the level of heme d at the interface between CydA and CydB. The E74F mutation results in a labile oxidase that cannot be purified by chromatographic techniques. However, the UV-vis spectrum of a detergent extract from the mutant clearly shows that the absorbancies of heme d are fully lacking, while some residual absorbance of heme b_{595} is detectable (shown in Supplementary Figure 7). Thus, this new mutant underlines our proposed model. A summary of this discussion is now also provided in the legend to Figure 3.

We would like to thank both reviewers for the very helpful and constructive comments.